# App to support Recovery in Early Intervention Services (ARIES) study: protocol of a feasibility randomised controlled trial of a self-management Smartphone application for psychosis

Thomas Steare,[1,2] Puffin O'Hanlon,[1] Michelle Eskinazi,[1] David Osborn,[1,2] Brynmor Lloyd-Evans,[1,2] Rebecca Jones,[1] Helen Rostill,[3] Sarah Amani,[4] Sonia Johnson[1,2]

¹Division of Psychiatry, University College London, London, UK
²Camden and Islington NHS Foundation Trust, London, UK
³Surrey and Borders Partnership NHS Foundation Trust, Leatherhead, UK
⁴Early Intervention in Psychosis Programme (South of England), NHS England, Oxford, UK

**Correspondence to**
Professor Sonia Johnson;
s.johnson@ucl.ac.uk

## ABSTRACT

**Introduction** Mental health interventions delivered through digital technology have potential applications in promoting recovery and improving outcomes among people in the early stages of psychosis. Self-management approaches are recommended for the treatment of psychosis and could be delivered via applications (apps) installed on Smartphones to provide low-cost accessible support. We describe the protocol for a feasibility trial investigating a self-management Smartphone app intervention for adults using Early Intervention in Psychosis (EIP) services.

**Methods and analysis** In this feasibility randomised controlled trial, 40 participants will be recruited from EIP services in London and Surrey. Twenty participants will be randomised to receive a supported self-management Smartphone app (My Journey 3) plus Treatment As Usual (TAU), while the other 20 participants will receive TAU only. The primary objective of this study is to evaluate the feasibility of conducting a full-scale trial of this intervention in EIP services. Participant data will be collected at baseline and at two follow-up assessments conducted 4 months and 12 months post-baseline. Analysed outcome measures will include relapse of psychosis (operationalised as admission to a hospital or community acute alternative), mental health and well-being, recovery, quality of life and psychopathology. Semi-structured interviews with participants and EIP service clinicians will additionally explore experiences of using My Journey 3 and participating in the trial and suggestions for improving the intervention.

**Ethics and dissemination** The App to support Recovery in Early Intervention Services study has been reviewed and approved by the National Research Ethics Service Committee London—Brent (Research Ethics Committee reference: 15/LO/1453). The findings of this study will be disseminated through peer-reviewed scientific journals and conferences, magazines and web publications.

**Trial registration number** ISRCTN10004994.

## Strengths and limitations of this study

► The app has been well developed prior to the feasibility trial, including substantial stakeholder input to the development of the intervention at all stages, an initial period of testing in clinical settings, and refinement of the current version through lab and field testing.
► The acceptability and feasibility of the intervention and the study design of a future full-scale trial will be explored using a variety of measures, including recruitment and retention rates, usage data, and qualitative interviews on experience of use.
► The trial will only recruit Android Smartphone users as My Journey 3 is not as yet compatible with other Smartphone operating systems; this will limit sample representativeness.
► This is a feasibility study, thus not powered to draw conclusions regarding the effectiveness of the intervention.

society-level.[1–3] The implementation of Early Intervention in Psychosis (EIP) services across the UK has been associated with improved outcomes,[4 5] however, substantial challenges remain. Long-term follow-up studies have shown that relapse rates in the early course of psychosis remain high.[6] Furthermore functional recovery is often not attained following treatment.[7]

Thus, there is a need to increase and sustain benefits from EIP services. An approach with demonstrated efficacy in improving functional recovery and reducing relapse in psychosis is supported self-management.[8] Self-management has been designed to empower people as active agents in their own recovery by enabling them to develop skills such as recognising and monitoring

## INTRODUCTION

Psychosis is associated with significant costs at individual-level, family-level and

symptoms and early warning signs of relapse, identifying and avoiding stressors, and using effective coping strategies.[9] Self-management is associated with significant benefits including reduced distress, improved medication adherence and reduced number of hospitalisations.[10–12] Relapse prevention work has long been proposed as a key element in EIP,[13] and establishing and working towards recovery goals is widely advocated in EIP services.[14] However, well-evaluated tools and methods are currently not available to support widespread implementation of self-management approaches in EIP services.

Over recent years there has been growing interest in the development of innovative technologies in healthcare due to their potential to improve accessibility, efficacy, quality and cost-effectiveness of treatment.[15] Due to rapid advancements in mobile phone technology, it is possible to deliver clinical interventions via applications (apps) installed on Smartphones (mobile phones with computational capacities). Smartphones are widely available in the UK with over 75% of adults owning one.[16] They are often carried on the person with high accessibility to the internet and are therefore a suitable device to provide time-unlimited interventions in almost any location.[17]

Recent evidence suggests that people with psychosis are adopting digital technology in a similar way to the general population,[18–20] and that they are interested in using mental health interventions delivered via Smartphones.[21 22] A systematic review has found that interventions delivered through apps or text messaging on Smartphones are acceptable and feasible for people with psychosis and may support recovery.[23] Smartphone apps based on self-management principles have shown promise in a wide range of long-term health conditions,[24] and are a potential way to deliver accessible low-cost support to adults with psychosis.

Although emerging research suggests that Smartphone apps hold promise in delivering effective interventions to adults with psychosis, the evidence base is under-developed in comparison to more common mental disorders.[25] To date only one randomised controlled trial (RCT) of a psychosocial Smartphone app used in EIP services has been published.[26] In this proof-of-concept trial, 36 adults accessing EIP services were randomised to use a cognitive-behavioural therapy informed Smartphone app (Actissist) that aims to encourage active self-management, or a symptom-monitoring app which was classed as the control condition. Actissist was found to be acceptable, feasible and safe for adults accessing EIP services. Although not powered to find an effect, the study suggests that Actissist may confer benefits to users' outcomes over and above a passive symptom-monitoring app. Participants however only had access to the app for 12 weeks, were followed-up over a relatively short time frame (22 weeks) and significant mental health outcomes such as relapse were not measured. This work has been conducted in parallel with our study, and we anticipate both programmes of work will contribute to developing an evidence base as to whether Smartphone apps that promote self-management can improve outcomes in first-episode psychosis and reduce healthcare costs.

## Aims and objectives

The aim of the App to support Recovery in Early Intervention Services study is to examine the feasibility of conducting a full-scale trial of a clinician supported self-management Smartphone app for adults accessing EIP services. The feasibility trial will aim:

1. To identify whether a self-management Smartphone app is acceptable to use and feasible to support in EIP services in the context of a research study, and to identify any necessary modifications to the intervention content and design, or to its delivery in EIP services.
2. To test the feasibility and acceptability of trial procedures for a definitive trial.
3. To test procedures for evaluating intervention engagement and participant outcomes.

## METHODS AND ANALYSIS
### Design

The study is a feasibility RCT comparing a clinician supported self-management Smartphone app (My Journey 3), in addition to Treatment As Usual (TAU), to a control group receiving TAU only. The design as described here adheres to the Standard Protocol Items: Recommendations for Interventional Trials (SPIRIT).[27] A copy of the SPIRIT checklist is provided as online supplementary file 1. Relevant items from the WHO Trial Registration Data Set are detailed in online supplementary file 2.

### Setting

The trial will take place in six EIP services across three National Health Service (NHS) Foundation Trusts in England: Camden and Islington; East London; and Surrey and Borders Partnership. All participating EIP services provide care co-ordination to service users, access to a psychiatrist and psychiatric medication and psychosocial interventions, aiming to conform to the current UK model for EIP services.[28] None of the participating EIP services offer Smartphone apps or any other digital interventions as part of routine care, nor are structured self-management tools consistently available in these teams. A list of participating sites is available from the authors.

### Participants

Forty service users will be recruited from the participating EIP services. Assuming a conservative estimate of a 40% attrition rate, the study should feature 12 completer participants in each group as recommended for feasibility and pilot trials.[29] The proposed sample size will therefore be sufficient to establish the acceptability of My Journey 3 and the feasibility of trial procedures.

### Eligibility criteria
#### Inclusion criteria
- Aged 16 or older.
- Have experienced at least one episode of psychosis.
- Currently on the caseload of an EIP service and in contact with clinicians.

► User of a Smartphone with an Android operating system.

### Exclusion criteria

► Lack of capacity to provide informed consent to participate in the trial.
► Inability to communicate and understand English sufficiently to understand trial procedures and use My Journey 3.
► In the view of their EIP service, poses such a high risk to others that it would be unsafe to conduct research meetings even on NHS premises.

Criteria are deliberately broad in order to reach conclusions generalisable to a wide range of EIP service users. Due to limited resources My Journey 3 has been developed for one Smartphone operating system (Android) at this stage of testing. Nearly half of Smartphones used in the UK use Android operating systems.[16] There is also emerging evidence that Android may be the Smartphone operating system most often used by adults with severe mental illness.[21] Resources are not currently available to deliver My Journey 3 in languages other than English.

### Recruitment

An overview of the recruitment procedure is displayed in the flow diagram (figure 1). Clinicians will identify EIP service users who appear potentially eligible for the trial. They will then make initial contact, explain the trial and ask if the potential participant is willing to be contacted by a researcher. If a potential participant is eligible and interested in taking part, a researcher will contact them to further explain the trial and to arrange a face-to-face meeting. At this meeting the researcher will provide the trial information sheet, confirm eligibility, invite any questions about participating and assess the service user's capacity to provide informed consent. Written consent to participate will be obtained using a consent form (online supplementary file 3) prior to undertaking the baseline assessment at the same meeting. Participants will be notified that they will be free to withdraw from the study at any time-point. Prior to study recruitment, researchers will undertake training in how to assess mental capacity and take informed consent.

To minimise loss to follow-up, participants will be asked to give their preferred contact details, and for their permission for the researcher to contact a nominated close other or EIP service clinician whom could contact the participant if the researcher is unable to contact the participant directly.

### Randomisation

At a separate time-point following the baseline assessment meeting participants will be randomly allocated in a 1:1 ratio to an intervention group where they will have access to My Journey 3 during the trial (n=20) or to a

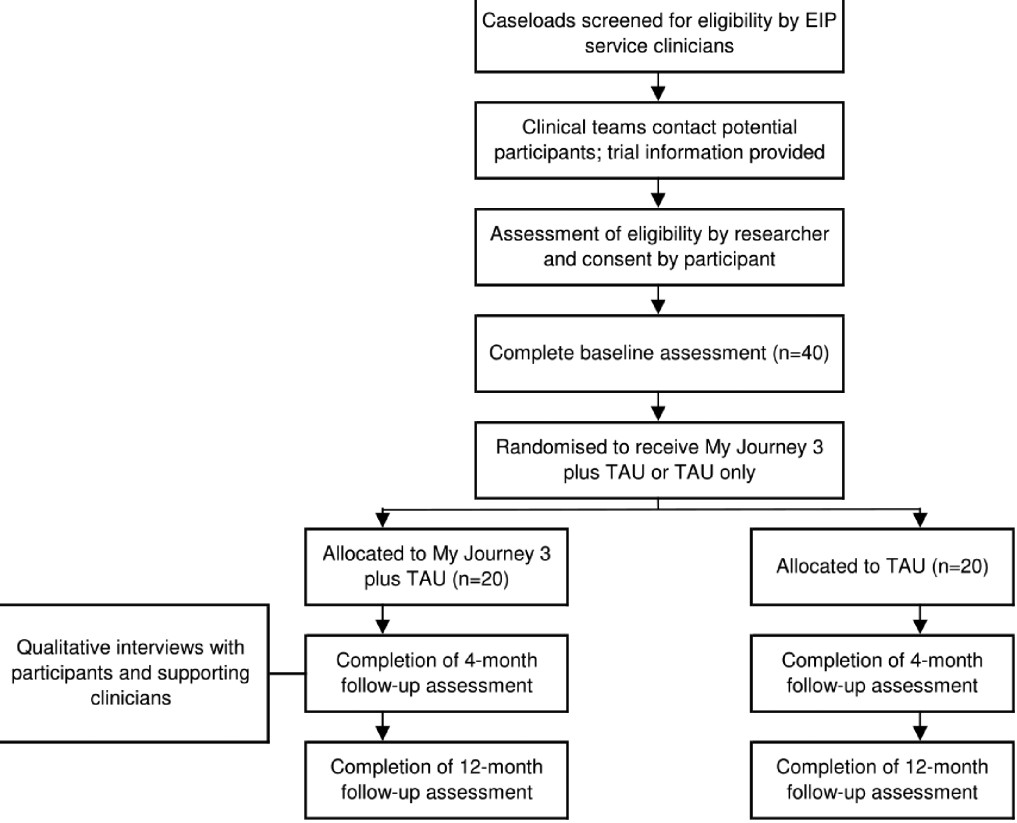

**Figure 1** SPIRIT flow diagram outlining the phases of the App to support Recovery in Early Intervention Services feasibility trial. EIP, Early Intervention in Psychosis; SPIRIT, Standard Protocol Items: Recommendations for Interventional Trials; TAU, Treatment As Usual.

control group (n=20). Randomisation will be conducted by an independent statistician. The allocation sequence will be concealed from the researcher, who will be blind when recruiting participants and conducting baseline assessments. An independent researcher will hold on to the allocation list and disclose participants' allocation to the trial researcher after the completion of the baseline assessments. Participants will then be informed of their allocation by the trial researcher. Participants cannot be blinded to their group allocation given the nature of the trial intervention and control group. As a single researcher will carry out most of the data collection, it is not practical for the group allocation of participants to be concealed from the research team in this feasibility study.

### The intervention
#### Development
My Journey 3 has been developed through a series of iterations. The first version of My Journey was designed by Surrey and Borders Partnership NHS Foundation Trust (led by Sarah Amani), after local EIP service users consulted about their care suggested there would be significant benefits from a Smartphone app that could be used for appointment and medication reminders, to track their mood and to share their recovery progress with EIP staff or carers. In 2011 a project group consisting of EIP service users, NHS clinicians, a pharmacist and an NHS manager was formed to drive the development of the My Journey app. The following year a prototype of My Journey was tested by 20 EIP service users to further inform development of the app. Since April 2013 this version of My Journey has been available for download from Google Play (the official app store for Android Smartphones).

The original version of My Journey contains generic advice on what to do if certain clinical difficulties arise, but does not allow personalised relapse prevention planning or recovery goal tracking. User feedback suggested that this was a limitation. In the current study we have collaborated with digital health experts, EIP service clinicians and adults with lived experience of psychosis to adapt existing paper-and-pen self-management intervention components—in routine use in NHS services—to be suitable for delivery in an app format.[30 31] In creating the product specification of My Journey 3 we have incorporated these self-management components with functions from the original My Journey. Technical development of My Journey 3 has been led by MyOxygen (https://myoxygen.uk/), a private app development company based in the UK.

Following the initial development of My Journey 3, it has been refined in response to two phases of preliminary testing. In the first phase six volunteer EIP services users participated in usability lab tests with My Journey 3 on their own Android Smartphone. Usability testing featured 'think aloud' tests where participants completed set tasks using My Journey 3 while providing a continuous commentary on their thoughts. This method was used to highlight design and usability issues and users' immediate reactions to the app. Individual interviews were also conducted to explore participants' perceptions of the ease of use of My Journey 3, concerns they might have and suggestions for improvements. My Journey 3 was modified to reflect the findings at this stage.

In the second phase a further six volunteer service users trialled My Journey 3 on their own Android Smartphone during a 1-month field study, with EIP service clinicians asked to support their clients' use of the app during routine appointments. Following the field study, individual interviews were conducted with service user participants to explore how they used My Journey 3 and how it could be improved. Further interviews with EIP service clinicians explored their experience of supporting their client with My Journey 3. Based on these findings My Journey 3 was updated again prior to the feasibility trial.

#### Intervention outline
My Journey 3 is a Smartphone app that has been designed to be used alongside EIP service care and with the support of clinicians. My Journey 3 aims to develop and support users' illness self-management skills to help facilitate recovery from first-episode psychosis. The My Journey 3 home screen can be seen in figure 2.

Three main components of My Journey 3 have been directly taken from the original app and updated. Information regarding psychosis, mental health and mental health services is provided in My Journey 3 through links to relevant NHS and voluntary sector websites and through videos of personal recovery stories. To facilitate symptom recognition and monitoring My Journey 3 also features a self-monitoring tool and symptom tracker where users can monitor 14 different symptoms and three lifestyle behaviours. Advice to help manage symptoms is provided after completing the symptom tracker, and a graphical summary of symptom severity over time is displayed. Users also have access to a pill tracker where they can log whether they have taken their psychiatric medication. The pill tracker features a daily alert at a pre-set time to remind users to input if they have taken their medication.

In designing My Journey 3 we have also drawn on evidenced-based self-management interventions to add structured intervention components focused on recovery planning and relapse prevention.[30 31] These additions allow for users to interactively:
► Identify strategies and coping resources that they find useful in maintaining well-being.
► Set and track progress towards personal recovery goals.
► Identify personal early warning signs of relapse and strategies and coping mechanisms to put in place should they experience these.
► Create a 'relapse plan', an action plan to follow in times of crisis in order to avoid or attenuate relapse.

My Journey 3 has been designed to be used by EIP service users in collaboration with clinicians. Clinicians

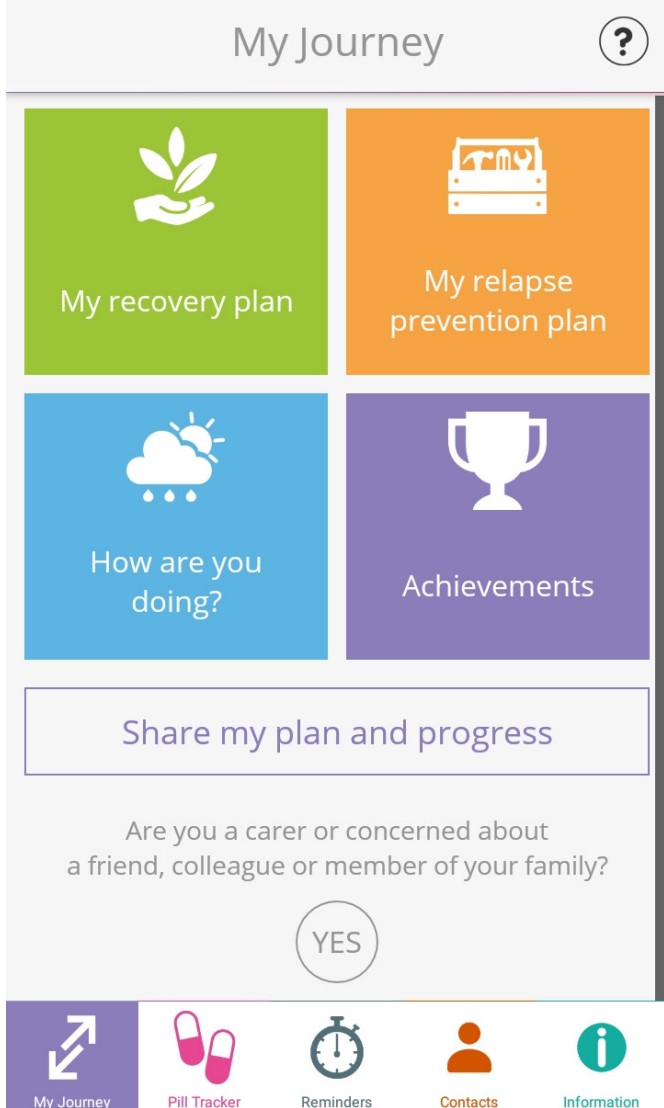

**Figure 2** The My Journey 3 home screen, which is seen by the user when accessing the app on their Smartphone.

can input relevant information to sections such as the relapse prevention plans and, with training, can provide assistance with the app. It is also suitable for independent use: the developers' aspiration is for the app to be initially used in collaboration with clinicians, but for it then to support users' self-management following discharge from EIP services.

My Journey 3 features weekly discrete notifications that appear on the users' Smartphone interface to encourage engagement with the app. Users also have the opportunity to set activity reminders that notify the user to engage in coping strategies and take part in pre-identified activities to promote well-being.

My Journey 3 also features a sharing functionality for users who wish to share their data with clinicians, family members, friends or other trusted third parties. This uses built-in sharing functionality of the user's Smartphone, such as e-mail. Participants will have control of who they choose to share their data with at all times.

### Delivery of the intervention

Each participant in the intervention group will take part in an individual training session with their supporting EIP service clinician and a researcher within 6 weeks of the initial consent meeting. During each training session participants will download My Journey 3 on to their Smartphone. The researcher will then give a demonstration of the app and its main functions. Participants will have the opportunity to practice using My Journey 3 and to ask questions. To facilitate use during the trial participants will be asked to input appropriate information in to the main functions of My Journey 3.

Participants will have access to My Journey 3 from the training session till the 12-month time-point. Participants in the intervention group will be free to withdraw from using My Journey 3, or decline the installation of it on to their Smartphone, without any impact on their study participation or their clinical care.

In line with evidence that clinician involvement increases user engagement with Smartphone apps,[32] participants' main contact for help with using My Journey 3 and the various intervention components will be their supporting EIP service clinician. Supporting EIP service clinicians will be encouraged by the researcher to discuss recovery goals and relapse prevention plans with study participants in routine appointments and then to assist participants in inputting them into My Journey 3. Clinicians will also be encouraged to regularly check with participants if they have been using My Journey 3 and if they need any further support with it. Clinician support will not be manualised. If supporting clinicians leave the participants' EIP service the researcher will arrange a meeting to introduce any new clinicians that have clients using My Journey 3, to the intervention and how they can support service users that have access to the app. Researcher support with My Journey 3 will be limited to the initial installation and technical support as needed during the trial period.

The research team will have no responsibility for providing clinical care if any My Journey 3 data shared by a participant to a third party indicates a decline in their mental health. If any major concerns regarding participants' well-being such as suicidality arise from My Journey 3 data shared by a participant to the researcher, this would be communicated to the EIP service or other appropriate mental health services. Third parties such as EIP service clinicians and carers will not be briefed by the research team on how to respond to such information but would be expected to act as appropriate in such event. Participants will be informed at the training session that any data that may suggest a decline in their mental state that they have shared with EIP service clinicians or carers may be acted on accordingly, but that My Journey 3 is not suitable for seeking urgent medical care while in crisis.

### Control group

Participants in the control group (n=20) will receive TAU that will be unaffected by their participation.

**Table 1** Timeline of participant enrolment, interventions, assessments and patient records data collection

| Time point | Enrolment | Baseline | Allocation | MJ3 training session* | 4-month follow-up | 12-month follow-up |
|---|---|---|---|---|---|---|
| **Enrolment** | | | | | | |
| Eligibility screen | X | | | | | |
| Informed consent | | X | | | | |
| Randomisation | | | X | | | |
| **Intervention** | | | | | | |
| My Journey 3 (intervention group) | | | | ←——————————→ | | |
| TAU (all participants) | | | | ←——————————→ | | |
| **Assessments** | | | | | | |
| PANSS | | X | | | X | X |
| Socio-demographic information | | X | | | X | X |
| Clinical service use | | X | | | X | X |
| Social Outcomes Index | | X | | | X | X |
| Mental Health Confidence Scale | | X | | | X | X |
| The Process of Recovery Questionnaire | | X | | | X | X |
| WEMWBS | | X | | | X | X |
| The DIALOG Scale | | X | | | X | X |
| Service Engagement Scale (completed by EIP service clinicians) | | X | | | | X |
| Qualitative interviews (with participants in the intervention group and supporting clinicians) | | | | | X | |
| **Patient records data (from previous 12 months to time point)** | | | | | | |
| Number of admissions to acute mental health services | | X | | | | X |
| Number of compulsory admissions to acute mental health services | | X | | | | X |
| Total number of days in acute care | | X | | | | X |
| Number of kept appointments with community mental health services | | X | | | | X |
| Number of missed appointments with community mental health services | | X | | | | X |
| Primary ICD-10 diagnosis | | X | | | | X |
| Most recent care cluster | | X | | | | X |
| Care Programme Approach status | | X | | | | X |

*Participants in the intervention group only.

EIP, Early Intervention in Psychosis; ICD-10, Internal Classification of Diseases: 10th Revision; MJ3, My Journey 3; WEMWBS, Warwick-Edinburgh Mental Well-being Scale; PANSS, Positive and Negative Syndrome Scale; TAU, Treatment As Usual.

## Treatment As Usual

TAU for service users attending EIP services typically involves regular meeting with a care co-ordinator, support from multi-disciplinary clinicians and access to a psychiatrist, psychiatric medication and a range of psychological interventions.

## Data collection

The participant timeline is summarised in table 1. All participants will be asked to complete self-report questionnaires during a structured assessment with a researcher at three time points: baseline (prior to randomisation), 4 months post baseline and 12 months post baseline. Participants in the intervention group will also be invited to complete an audio-recorded interview with a researcher during the 4-month follow-up assessments. Before arranging each assessment, the researcher will check with any clinicians in contact with the service users that the participant does not pose a risk to others and himself/herself and that it would be safe to conduct a research meeting. If there is an identified risk too serious for a meeting on NHS premises the participant will not be met for the assessment at that time-point, with the reason

for not arranging a meeting documented. If any major concerns regarding participants' risk or well-being arise during interactions with the researcher, this would be communicated to the appropriate EIP service.

Prior to the completion of each assessment participants will be provided with a trial information sheet. Their capacity to give informed consent will be assessed and their consent documented in writing. Participants will receive 20 pounds as a token of thanks for completing each assessment. Participants that have been discharged from EIP services during the trial will still be invited to attend assessments.

Once the recruitment target for the study has been met, the researcher will contact the appropriate administrators or informatics team within each NHS trust to arrange the collection of participant data from patient records. The researcher will also arrange for data to be collected from patient records 1 year later.

To test procedures for evaluating engagement with the intervention, data regarding My Journey 3 use will be collected throughout the trial period for all participants in the intervention group. My Journey 3 will automatically upload encrypted usage data to a secure trial server when the user has internet access on their Smartphone.

## Measures
### Data from patient records
The following data will be collected from patient records at baseline and the 12-month time-point:

a. Most recent clinical diagnosis as recorded in patient records using the Internal Classification of Diseases: 10th Revision classification.

b. Most recent care cluster (a classification of mental health service users based on their needs, used in the NHS).

c. Whether participants are subject to a care programme approach (NHS mental health services' case management model).

d. Service use during the previous 12 months measured at baseline and 12 months post study entry. Data will include history of use of acute mental health services, inpatient admissions and compulsory admissions and the number of kept and missed appointments with mental health services. Relapse of psychosis, the proposed primary outcome for a fully powered RCT, will be operationalised as participant admission to an acute mental health service (inpatient psychiatric wards, crisis houses, crisis resolution teams and acute day care services). This definition of relapse has been used previously in a recent trial of a self-management intervention in a mental health setting.[33]

### My Journey 3 usage data
The data collected will be a record of each time the user opens My Journey 3, whether this was in response to a prompt, and which components they use. Usage data will not include participants' text input or responses to self-rated questions, and will therefore not give any information regarding the mental health of users. Usage data will be solely used to assess the acceptability of My Journey 3 and user engagement with it.

### Self-report questionnaires
The following are measured as potential secondary outcomes for a future fully-powered RCT:

Service use measures over 1 year of follow-up.

1. Engagement with EIP services as measured by the Service Engagement Scale (SES),[34] a 14-item questionnaire that measures engagement for four different dimensions: availability, collaboration, help-seeking and treatment adherence. The SES will be completed by EIP service clinicians, such as participants' care co-ordinators, at baseline and at the 12-month time-point.

Measures at baseline, 4-month and 12-month follow-up assessments.

1. Psychotic symptoms and general psychopathology, measured by the Positive and Negative Syndrome Scale (PANSS), a 30-item scale that yields three separate scores on positive symptoms, negative symptoms and general psychopathology.[35] To inform the PANSS a trained researcher will conduct clinical interviews with participants at each assessment.

2. Social outcomes, rated by the Social Outcomes Index (SIX),[36] a 6-item index of social outcomes and circumstance.

3. Mental health-related self-efficacy, measured by the Mental Health Confidence Scale,[37] a 16-item self-report scale of service users' confidence in their ability to cope with stressful or difficult events.

4. Self-rated recovery measured by the Questionnaire about the Process of Recovery (QPR),[38] a 22-item measure, yielding a total score and subscale scores for intrapersonal and interpersonal recovery factors.

5. Mental well-being, rated by total score on The Warwick-Edinburgh Mental Well-Being Scale (WEMWBS),[39] a 14-item self-report scale of mental well-being.

6. Subjective quality of life and satisfaction with treatment measured by The DIALOG scale,[40] an 11-item self-report scale.

7. Socio-demographic characteristics including age, gender and ethnicity. We will also collect data regarding accommodation and living situation, employment status, educational attainment and Smartphone use, including use of other mental health apps.

### Qualitative interviews
Interviews with participants in the intervention group will follow a topic guide and will explore their experience of using My Journey 3, including:

► The usability and acceptability of My Journey 3.
► Positives and negative aspects of My Journey 3.
► Impact of My Journey 3 on their life.
► Facilitators and barriers to using My Journey 3.
► Views on the training session.

► Views on the support they received from their clinician in using My Journey 3.

EIP service clinicians who have been supporting participants with My Journey 3 will also be asked to complete an audio-recorded interview close to the time the participant completes the 4-month follow-up assessment. Written consent to take part will be confirmed beforehand, with an information sheet provided. Clinicians will be asked to provide demographic data prior to undertaking the interview. The interview will follow a topic guide and will focus on:

► Positives and negative aspects of My Journey 3.
► The experience of supporting clients with My Journey 3.
► Facilitators and barriers to providing support and to the incorporation of My Journey 3 in clinical management.
► Views on the training session.

## Data analysis

No pre-specified criteria have been set for establishing the acceptability of My Journey 3 or the feasibility of trial procedures. The acceptability of My Journey 3 for EIP service users and clinicians will be determined from feedback given from the qualitative interviews and the level of participants' My Journey 3 use indicated from the app usage data. Trial feasibility will be assessed from reviewing recruitment rates, drop-out rates and intervention enrolment and use during the trial period. These will be reviewed by the study team and discussed with stakeholders at a final dissemination event to decide on future steps to take.

### Quantitative analysis

We will report rates of recruitment and retention in the trial, and, for the intervention group, the level of usage of My Journey 3 during the trial. The demographic and clinical characteristics of participants at baseline will be summarised separately for each study group using descriptive statistics.

To pilot the methods of analysis for a full-scale trial we will use an intention-to-treat approach using data from all randomised participants. The effect of My Journey 3 on relapse (the proposed primary outcome for a fully-powered RCT) during the 12-month trial period will be estimated using logistic regression. The effect of the intervention on continuous outcome measures (SIX, QPR, WEMWBS, DIALOG scale, PANSS and SES) will be estimated using linear regression adjusting for the baseline measure of the outcome in question. Results will be summarised using effect estimates and 95% CIs only. No interim analyses are planned.

### Qualitative analysis

To assess the acceptability of My Journey 3 interview data with participants in the intervention group and supporting EIP service clinicians will be analysed based on The Theoretical Framework of Acceptability for healthcare system interventions.[41] Analyses will be conducted collaboratively by a group of researchers.

## Patient and public involvement

Independent advice will be sought from a trial steering group. Members will include researchers with expertise in developing and testing digital health interventions, EIP service clinicians, people with lived experience of first-episode psychosis and carers. Steering group members with lived experience of psychosis were consulted on the first paper prototypes of My Journey 3 and on the research protocol. The version of My Journey 3 tested in the current feasibility trial has been developed based on EIP service users' feedback from the usability tests and field study. Participants will be offered a summary of the research findings at the end of the study.

## Data monitoring and management

A secure password-protected trial database will be developed and managed to store all quantitative data using SPSS software V.23, and will feature non-identifiable trial IDs only. Data entry of participants' questionnaires will be primarily undertaken by the trial researcher, with a random sample checked by other research team members. Anonymised electronic interview transcripts will be checked for accuracy and stored using NVivo for Windows (QSR International Pty Ltd V.11, 2016). After the trial, all data will be archived securely at University College London.

Due to the small sample size of the study a data monitoring committee is not planned, but the steering group will advise if one is later needed.

## ETHICS AND DISSEMINATION
### Confidentiality

Participant data will be accessed by the research team only. Consent forms and data collection forms will be securely stored in locked cabinets at University College London. Data collection forms will not feature participants' names but a unique trial ID that could not be linked to participants by anyone outside the research team. Consent forms that identify participants will be kept separately from data collection forms.

Password-protected electronic data will be stored on the secure IT network at University College London. App usage data collected while using My Journey 3 will be anonymised and encrypted and will not contain personal user information.

### Serious adverse events

Serious adverse events such as hospital admissions and death reported to the trial team will be reviewed by the chief investigator. Identified adverse events assessed as trial-related will be reported to the trial sponsor.

## Dissemination

Results will be disseminated through scientific publications, and to a wider audience via magazines and web publications.

**Acknowledgements** The ARIES research team are grateful to their software collaborators MyOxygen for their technical development and hosting of My Journey 3 and to Ali Mousa for his valuable contribution to the development of the original My Journey app. We are grateful to Max Birchwood for his permission to incorporate 'Back in the Saddle' in to My Journey 3. We are grateful to Rachel Perkins for her permission to adapt the Personal Recovery Plan resource and incorporate in to My Journey 3.

**Contributors** The trial design was developed by SJ, DO, BL-E and POH. SA, HR, POH and ME have led on the development of the intervention. RJ has advised on the statistical analysis. SJ is the chief investigator, based at University College London, DO the co-chief investigator and TS the project manager. All authors have contributed and approved this manuscript.

**Funding** The research is funded by the National Institute for Health Research (NIHR) Collaboration for Leadership in Applied Health Research and Care North Thames at Barts Health NHS Trust (NIHR CLAHRC North Thames). SJ, DO and BL-E are supported by the NIHR Mental Health Research Policy Unit, the NIHR Collaboration for Leadership in Applied Health Research and Care (CLAHRC) North Thames and the UCLH Biomedical Research Centre. Trial Sponsor: Camden & Islington NHS Foundation Trust. sponsor.noclor@nhs.net

**Disclaimer** The views expressed in this article are those of the authors and not necessarily those of the NHS, the NIHR or the Department of Health and Social Care.

**Competing interests** None declared.

**Patient consent for publication** Not required.

**Ethics approval** Ethical approval has been obtained from the London Brent National Research Ethics Service Committee (Research Ethics Committee reference: 15/LO/1453) which has approved all amendments to protocol. Future protocol modifications will be submitted for approval to the research ethics committee and communicated to the study sponsor, site principal investigators, participating National Health Service trusts and participants. The current protocol in use is V.9, 29 July 2017.

**Provenance and peer review** Not commissioned; externally peer reviewed.

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
