## [Reviewer comments · BMJ Open]

ARTICLE DETAILS

TITLE (PROVISIONAL)	The App to support Recovery in Early Intervention Services (ARIES) Study: Protocol of a feasibility randomised controlled trial of a self-management Smartphone application for psychosis
AUTHORS	Stear, Thomas; O'Hanlon, Puffin; Eskinazi, Michelle; Osborn, David; Lloyd-Evans, Brynmor; Jones, Rebecca; Rostill, Helen; Amani, Sarah; Johnson, Sonia

VERSION 1 - REVIEW

REVIEWER	John Gleeson School of Behavioural and Health Sciences Australian Catholic University
REVIEW RETURNED	03-Sep-2018

GENERAL COMMENTS	This is a potentially important pilot study that builds upon at least 7 years of background development work. It is very pleasing to see the level of involvement of consumers to date in this project and the support from services in getting the project to this point. The rationale for the project is well justified and the four aims of the study are clearly outlined. If these four aims are appropriately addressed through the study methods, the results will have the potential to support a full-scale RCT of the intervention. However, the protocol could benefit from a number of clarifications so that the background, aims, measures, and planned analyses are appropriately aligned. I outline these suggestions with reference to the specific section of the SPIRIT checklist. Section 11 Interventions: It was not clear for the reader whether there was any back-end or researcher interface to the My Journey APP. If so, what would be in place by way of monitoring the use of the App through the course of pilot, and would there be any data entered into the APP by users that could indicate a change in mental health (e.g., change in mood) and if so is there a safety protocol for responding to such changes? Related to this, are there any principles or protocols for maximizing engagement in the APP? What will happen if a participant completely disengaged from the APP? In addition, what kind of IT support will be provided to users throughout the pilot? If users share their data with other (e.g., carers), what will happen if these data raise concerns regarding the participant's well-being? Will there be any criteria for withdrawal from the APP or from the study? If a relapse occurs what is the plan for user involvement or if risk status changes during the course of the study? Are participants free to use any other mental health APPS during the pilot?
---

	Section 12 Outcomes. Although it is not mentioned in the abstract, the data analyses section outlines that the effect of the APP on relapse will be estimated. However, there is no detail as to how relapse will be determined in individual cases. Obviously there is a risk of bias for this specific outcome so thought should be given as to how this is managed. Section 13 Time Schedule: What are the planned dates for the study? Section 14 Sample Size: If one of the aims is to establish the feasibility of a larger trial that will have the aim of assessing the effectiveness of the APP in preventing relapse, has it been determined that the sample size large enough to assess this? In making this judgment have the team factored in the likely rate of relapse, and the rate of attrition from the study and rate of non-participation in the APP? Section 16 Sequence Explain why stratification is not needed given that participants are being recruited across multiple sites. Section 18. Given that this is a 12-month follow up study, please explain strategies for retention of participants in the study. If participants are discharged from their EIP service during the course of the study will this affect their participation in the study? If clinicians resign from their post during the course of the trial is there a strategy for addressing this? Section 20: Statistical Methods: My major concern was that it was not clear how the four specific aims were going to addressed though the analyses. Did the researchers give consideration to specifying criteria in relation to domains of acceptability and feasibility, which are referenced in the four aims? For example, what will be the criteria that the team will use to determine if the APP is acceptable and/or feasible? Section 21 Monitoring: Will there be any stop/amend/go criteria for the trial? Section 24 Research Ethics and Approval: If the team has not already done so, I would advise checking with the MHRA as to whether the APP needs to be registered as a medical device.
--	---

REVIEWER	Angus MacBeth University of Edinburgh Scotland UK
REVIEW RETURNED	04-Sep-2018

GENERAL COMMENTS	This is an interesting and timely protocol, given the increasing interest in mobile and app-assisted delivery of support for mental health difficulties in general, including psychosis. I have two minor queries: 1. The authors note that only one previous RCT in the area has been published. Could they clarify how the current study differs and or builds on the previous trial?
---

	2. Presumably TAU includes the possibility that participants in this condition can download other apps for mental health support? Will this be captured in any way?
--	---

REVIEWER	Susanne Kraft District Hospital Guenzburg, Department of Psychiatry II, Ulm University, Germany
REVIEW RETURNED	14-Sep-2018

GENERAL COMMENTS	This very well written study protocol describes the design of an on-going feasibility randomized controlled trial. The analysed intervention – a self-management smartphone app to support recovery in people with psychosis – is, to my opinion, highly promising as it provides a novel and sophisticated intervention which could be implemented in daily care at low-cost, if feasible. I have some questions and suggestions for a minor revision: a) Abstract – Methods and analysis: Sentence „Outcome measures will include“: I suggest to start the sentence with „Analysed outcome measures ...“, as these measures are not the outcomes of this feasibility trial, but will be analysed regarding their usability in a future full scale trial. b) Strength and limitations of this study: The last sentence „Blinding has not been feasible in this trial“ reads as a study result. As this is a study protocol, no results should be presented. I suggest to omit or rephrase this sentence. c) Aims and objectives: Please clarify the differences between aim #2 and aim #4. Otherwise, merge into one single aim. Explain in some more detail. To my understanding, recruitment and retention rates are indicators of the feasibility and acceptability of the trial procedures. d) Recruitment: Do informed consent, baseline assessment and randomisation all three take place at the same meeting with the participant? If not, please describe. e) Recruitment: You describe that the service user's capacity to provide informed consent is assessed by a researcher. Has he/she been trained before, and/or experienced? Please describe. f) Randomisation: Please describe the method of concealment (e.g. closed envelopes). g) The Intervention – Development: I suggest you include the information about the software collaborators/developers in this paragraph, and if it is a private company or not. h) The Intervention – Delivery of the intervention: If I understand correctly, the download and the introduction of the app takes place up to 6 weeks after randomisation. I suggest to include the information in the first row of Table 1, e.g. by adding a line „introduction to app“, to indicate that the intervention does not immediately start at the time of allocation. i) Data collection / Outcomes: I understand that you first describe the type of assessment (in the section „Data collection“) and then, in the outcomes section, the instruments used. This makes sense, but also leads to some redundancies, e.g. the paragraph „Data collection - Data from patient records“ and „Outcomes – further measures ...“, part c., first sentence. It might be easier for the reader if you merge large parts of the „Data collection“ and the „Outcomes“ section and use a new heading, e.g. „Measures“. I suggest the following structure:
--

	1) Data collection (until now: sub-section „Baseline and follow-up assessments“, but omit this sub-heading) 2) Measures (with the sub-sections: (2a) Data from patient records, (2b) My Journey usage data (2c) Self-report questionnaires and clinical interviews (until now: first part of „Outcomes“), (2d) Qualitative interviews. If you follow this suggestion, I propose j) – n): j) Data collection – Qualitative interviews: see i). Please move the first sentence to the „Data collection“ section. k) Data collection – Data from patient records: see i). Please move the first and second sentence („Once the ... year later.“) to the „Data collection“ section. Regarding the primary outcome: Did you think about using the number of days until relapse instead of an dichotomous measures (relapse yes/no)? If yes, please describe, also in the data analysis section. l) Data collection – My Journey usage data: see i). Please move the first sentence („Data regarding ...“) to the „Data collection“ section. I suggest to start the „My Journey ...“-sub-section with the sentence „To test procedures for evaluating intervention engagement, data regarding ...“. m) Outcomes: see i). I suggest to move the first sentence („The feasibility trial ...“) to the „Data analysis“ section or omit it. You could remove the second sentence („participant outcome data ...“), as this was already described in the „Data collection“ section. Please move the second paragraph („The proposed primary outcomeservices“) to the „Data from patient records“ section, rephrase if necessary. n) Outcomes – Further measures ... , part b. and c.: see i) Please move this information to the section „Data from patient records“, rephrase if necessary. o) Outcomes – Further measures ... , part c., last sentence: Please shortly explain „most recent care cluster“ and „care programme approach status“. p) Do you also collect the information on the time span needed for the assessments (questionnaires, interview)? If so, please state.
--	---

VERSION 1 – AUTHOR RESPONSE

Reviewer(s)' Comments to Author:

Reviewer: 1

Reviewer Name: John Gleeson

Institution and Country: School of Behavioural and Health Sciences, Australian Catholic University

Please state any competing interests or state 'None declared': None

Please leave your comments for the authors below This is a potentially important pilot study that builds upon at least 7 years of background development work. It is very pleasing to see the level of involvement of consumers to date in this project and the support from services in getting the project to this point.

The rationale for the project is well justified and the four aims of the study are clearly outlined. If these four aims are appropriately addressed through the study methods, the results will have the potential to support a full-scale RCT of the intervention.

However, the protocol could benefit from a number of clarifications so that the background, aims, measures, and planned analyses are appropriately aligned. I outline these suggestions with reference to the specific section of the SPIRIT checklist.

Section 11 Interventions: It was not clear for the reader whether there was any back-end or researcher interface to the My Journey APP. If so, what would be in place by way of monitoring the use of the App through the course of pilot, and would there be any data entered into the APP by users that could indicate a change in mental health (e.g., change in mood) and if so is there a safety protocol for responding to such changes?

Our response: We have added a statement explaining that researcher involvement with the intervention will be limited to the initial set-up and technical support only (page 14). The research team will have no access to any app use data that could indicate changes in participants' mental health, therefore we have no safety protocols regarding this.

Related to this, are there any principles or protocols for maximizing engagement in the APP?

Our response: There are no protocols to encourage engagement with the intervention. Thank you for this useful suggestion, which we will consider in a future definitive trial.

What will happen if a participant completely disengaged from the APP?

Our response: As suggested by the reviewer we have added a statement stating that participants are free to decline the intervention or stop using it (page 14).

In addition, what kind of IT support will be provided to users throughout the pilot?

Our response: We have added a statement explaining that researcher involvement with the intervention will be limited to the initial set-up and technical support only (page 14).

If users share their data with other (e.g., carers), what will happen if these data raise concerns regarding the participant's well-being?

Our response: We have added two sentences in the last paragraph of the 'Intervention outline' section (page 13), explaining the expected course of action if any My Journey 3 data shared to third parties raises concerns regarding participants' mental health. A sentence has also been added to the end of the first paragraph of the 'Data collection' section (page 15), explaining that any major participant risk identified by the researcher would be fed-back to the appropriate EIP service.

Will there be any criteria for withdrawal from the APP or from the study?

Our response: As suggested we have added details stating that participants are free to withdraw the intervention in the 'Data collection' section (page 14). We have not defined any criteria for withdrawal from the intervention. We will consider this helpful suggestion when planning a future full-scale trial.

If a relapse occurs what is the plan for user involvement or if risk status changes during the course of the study?

Our response: We have added a statement to explain that the researcher will check that participants' risk to others remains low before arranging research assessments (page 15). Potential relapse would not affect participation unless it has an impact on participants' capacity to give informed consent.

Are participants free to use any other mental health APPS during the pilot?

Our response: Participants will be free to use other mental health apps during the study. We have added details in how we will measure participants' use of other apps during the study (page 20).

Section 12 Outcomes. Although it is not mentioned in the abstract, the data analyses section outlines that the effect of the APP on relapse will be estimated. However, there is no detail as to how relapse will be determined in individual cases. Obviously there is a risk of bias for this specific outcome so thought should be given as to how this is managed.

Our response: As suggested we have added a statement on the study definition of relapse, in the 'Measures' section of the manuscript (page 18).

Section 13 Time Schedule: What are the planned dates for the study?

Our response: The date of first participant enrollment, and the current progress of the study, is provided already in the WHO Trial Registration Data Set (Additional file 2).

Section 14 Sample Size: If one of the aims is to establish the feasibility of a larger trial that will have the aim of assessing the effectiveness of the APP in preventing relapse, has it been determined that the sample size large enough to assess this? In making this judgment have the team factored in the likely rate of relapse, and the rate of attrition from the study and rate of non-participation in the APP?

Our response: We have added further details on how the sample size has been determined in the 'Participants' section (page 8). We have not considered rate of relapse in determining the appropriate sample size, however we would take into account evidence on relapse rates if considering selecting this as a primary outcome in a definitive trial. We are also not aware of the likely rate of participation in the app, therefore this was not considered when determining the sample size. The feasibility study will aim to assess levels of engagement with the app, which could then help inform the sample size of a larger definitive trial.

Section 16 Sequence Explain why stratification is not needed given that participants are being recruited across multiple sites.

Our response: The study is a feasibility trial which is under-powered by design. Statistical analyses will not have the power to detect differences in outcomes between the two groups. Stratification of participants is therefore not needed.

Section 18. Given that this is a 12-month follow up study, please explain strategies for retention of participants in the study.

Our response: A paragraph on how loss to follow-up will be minimized has been added (page 10).

If participants are discharged from their EIP service during the course of the study will this affect their participation in the study?

Our response: We have added details on how participants that have been discharged from EIP services during the study will still be invited to attend the study research assessments (page 15).

If clinicians resign from their post during the course of the trial is there a strategy for addressing this?

Our response: We have added details in the 'Delivery of the intervention' section explaining that the researcher will contact EIP service clinicians who have replaced resigned clinicians that have been supporting service users with My Journey 3, and introduce them to the app (page 14).

Section 20: Statistical Methods: My major concern was that it was not clear how the four specific aims were going to be addressed through the analyses. Did the researchers give consideration to specifying criteria in relation to domains of acceptability and feasibility, which are referenced in the four aims? For example, what will be the criteria that the team will use to determine if the APP is acceptable and/or feasible?

Our response: We have devised no pre-set criteria for establishing intervention acceptability or trial feasibility. Instead we will look at interview feedback and app usage data to see if participants and clinicians find the app acceptable and will look at app usage data to see how often the app is used. Trial feasibility will be assessed from reviewing recruitment rates, drop-out rates and intervention enrolment and use during the trial period. We have added a paragraph outlining this at the start of the data analysis section of the paper (page 24).

Section 21 Monitoring: Will there be any stop/amend/go criteria for the trial?

Our response: Our study is a feasibility trial, rather than an internal pilot. It does not have the immediate capacity to progress to a fully-powered RCT so there are no stop/amend/go criteria as a result.

Section 24 Research Ethics and Approval: If the team has not already done so, I would advise checking with the MHRA as to whether the APP needs to be registered as a medical device.

Our response: My Journey 3 does not meet the classification of a medical device by the MHRA (2014) This was discussed in detail with experts in app development within and outside the team during the initial development of the intervention. It is used to collect and store data rather than to make a diagnosis or prompt help seeking.

Reviewer: 2

Reviewer Name: Angus MacBeth

Institution and Country: University of Edinburgh, Scotland, UK

Please state any competing interests or state 'None declared': None declared

Please leave your comments for the authors below This is an interesting and timely protocol, given the increasing interest in mobile and app-assisted delivery of support for mental health difficulties in general, including psychosis. I have two minor queries:

1. The authors note that only one previous RCT in the area has been published. Could they clarify how the current study differs and or builds on the previous trial?

Our response: We have added further details about the previous RCT in the last paragraph of the introduction (page 6).

2. Presumably TAU includes the possibility that participants in this condition can download other apps for mental health support? Will this be captured in any way?

Our response: Participants will be free to use other mental health apps during the study. We have added details in how we will measure participants' use of other apps during the participant assessments (page 21).

Reviewer: 3

Reviewer Name: Susanne Kraft

Institution and Country: District Hospital Guenzburg, Department of Psychiatry II, Ulm University, Germany

Please state any competing interests or state 'None declared': None declared.

Please leave your comments for the authors below This very well written study protocol describes the design of an on-going feasibility randomized controlled trial. The analysed intervention – a self-management smartphone app to support recovery in people with psychosis – is, to my opinion, highly promising as it provides a novel and sophisticated intervention which could be implemented in daily care at low-cost, if feasible.

I have some questions and suggestions for a minor revision:

a) Abstract – Methods and analysis: Sentence „Outcome measures will include“: I suggest to start the sentence with „Analysed outcome measures ...“, as these measures are not the outcomes of this feasibility trial, but will be analysed regarding their usability in a future full scale trial.

Our response: Thank you for this suggestion. We have made this change in the abstract as suggested (page 3).

b) Strength and limitations of this study: The last sentence „Blinding has not been feasible in this trial“ reads as a study result. As this is a study protocol, no results should be presented. I suggest to omit or rephrase this sentence.

Our response: The suggested correction has been made in the strengths and limitations section (page 4).

c) Aims and objectives: Please clarify the differences between aim #2 and aim #4. Otherwise, merge into one single aim. Explain in some more detail. To my understanding, recruitment and retention rates are indicators of the feasibility and acceptability of the trial procedures.

Our response: The aims have been changed, with aim 4 omitted (page 3).

d) Recruitment: Do informed consent, baseline assessment and randomisation all three take place at the same meeting with the participant? If not, please describe.

Our response: We have clarified the timing of informed consent, baseline assessment and randomisation by rephrasing the 'recruitment' and 'randomisation' sections (pages 9 & 10).

e) Recruitment: You describe that the service user's capacity to provide informed consent is assessed by a researcher. Has he/she been trained before, and/or experienced? Please describe.

Our response: We have added a sentence regarding researcher training in the 'recruitment' section (page 9).

f) Randomisation: Please describe the method of concealment (e.g. closed envelopes).

Our response: A sentence regarding the method of allocation concealment has been added to the 'randomisation' section (page 10).

g) The Intervention – Development: I suggest you include the information about the software collaborators/developers in this paragraph, and if it is a private company or not.

Our response: As suggested we have added a sentence on our software collaborators stating that they are a private company based in the UK in the 'intervention development' section (page 11). We have also included an internet link to the company's website.

h) The Intervention – Delivery of the intervention: If I understand correctly, the download and the introduction of the app takes place up to 6 weeks after randomisation. I suggest to include the

information in the first row of Table 1, e.g. by adding a line „introduction to app“, to indicate that the intervention does not immediately start at the time of allocation.

Our response: The table has been updated, with an extra column added to indicate that for participants in the intervention group the intervention will start from when they attend the My Journey 3 training session (page 17).

i) Data collection / Outcomes: I understand that you first describe the type of assessment (in the section „Data collection“) and then, in the outcomes section, the instruments used. This makes sense, but also leads to some redundancies, e.g. the paragraph „Data collection - Data from patient records“ and „Outcomes – further measures ...“, part c., first sentence. It might be easier for the reader if you merge large parts of the „Data collection“ and the „Outcomes“ section and use a new heading, e.g. „Measures“. I suggest the following structure:

Our response: As suggested we have altered the structure of the data collection and outcomes section. The ‘Outcomes’ section has been replaced with ‘Measures’ (pages 18 to 22). This section now features four sub-headings in the following order: ‘Data from patient records’, ‘My Journey 3 usage data’, ‘Self-report questionnaires’ and ‘Qualitative interviews’.

1) Data collection (until now: sub-section „Baseline and follow-up assessments“, but omit this sub-heading)

Our response: The sub-heading ‘Baseline and follow-up assessments’ has been omitted.

2) Measures (with the sub-sections: (2a) Data from patient records, (2b) My Journey usage data (2c) Self-report questionnaires and clinical interviews (until now: first part of „Outcomes“), (2d) Qualitative interviews.

If you follow this suggestion, I propose j) – n):

j) Data collection – Qualitative interviews: see i). Please move the first sentence to the „Data collection“ section.

k) Data collection – Data from patient records: see i). Please move the first and second sentence („Once the ... year later.“) to the „Data collection“ section.

Our response: As suggested we have created a section called ‘Measures’, which features the four sub-headings ‘Data from patient records’, ‘My Journey 3 usage data’, ‘Self-report questionnaires’ and ‘Qualitative interviews’. The first sentence of the ‘qualitative interviews’ section has been moved to the first paragraph of the ‘data collection’ section and rephrased (page 15). The paragraph beginning ‘Once the recruitment target of the study has been met...’ has been moved to the ‘Data collection’ section and re-phrased (page 15).

Regarding the primary outcome: Did you think about using the number of days until relapse instead of an dichotomous measures (relapse yes/no)? If yes, please describe, also in the data analysis section.

Our response: Thank you for this useful suggestion, we will consider it in planning next steps with the evaluation of this app.

l) Data collection – My Journey usage data: see i). Please move the first sentence („Data regarding ...“) to the „Data collection“ section. I suggest to start the „My Journey ...“-sub-section with the sentence „To test procedures for evaluating intervention engagement, data regarding ...“.

Our response: The section on how My Journey 3 usage data will be collected has been moved to the ‘Data collection’ section as suggested. The text has been revised as suggested: ‘To test procedures

for evaluating engagement with the intervention, data regarding My Journey 3 use will be collected throughout the trial period for all participants in the intervention group..' (page 16).

m) Outcomes: see i). I suggest to move the first sentence („The feasibility trial ...“) to the „Data analysis“ section or omit it. You could remove the second sentence („participant outcome data ...“), as this was already described in the „Data collection“ section. Please move the second paragraph („The proposed primary outcome ...services“) to the „Data from patient records“ section, rephrase if necessary.

Our response: The paragraph ‘The feasibility trial is not powered to test hypotheses, but assess the feasibility of participant outcome measures for use in a future RCT. Participant outcome data will be collected from assessments with a researcher and from patient records’ has been omitted. The second paragraph (‘The proposed primary outcome ...services’) has been rephrased to describe the primary outcome more clearly and has been moved to the ‘Data from patient records’ section (page 19).

n) Outcomes – Further measures ... , part b. and c.: see i) Please move this information to the section „Data from patient records“, rephrase if necessary.

Our response: Further measures section ‘a’ has been moved to ‘Measures at baseline, 4-month and 12-month follow-up assessments’ (page 20). Further measures section ‘b’ and ‘c’ have been moved to ‘Data from patient records section’ (page 19).

o) Outcomes – Further measures ... , part c., last sentence: Please shortly explain „most recent care cluster“ and „care programme approach status“.

Our response: We have added definitions to explain ‘care clusters’ and ‘care programme approach’ in the ‘Data from patient records’ section (page 19).

p) Do you also collect the information on the time span needed for the assessments (questionnaires, interview)? If so, please state.

Our response: We will not collect any data on the time taken by participants to complete the research assessment meetings. We expect the research meetings to last for an hour, and the qualitative interviews to last for 20 minutes.

VERSION 2 – REVIEW

REVIEWER	John Gleeson Australian Catholic University
REVIEW RETURNED	12-Nov-2018

GENERAL COMMENTS	The authors have clarified a number of questions pertaining to the app and the measurement of outcomes which has improved the manuscript. I have some remaining issues that I believe require some further detail based on these initial clarifications. The authors have clarified that the researcher support will be limited to setting up participants on the app. Given the common problems of low engagement with mental health apps, could the authors please clarify their reasoning as to why this was considered sufficient for the purposes of the pilot study? In considering the response, please give consideration to the following recent review paper:
--

	Torous, J., Nicholas, J., Larsen, M. E., Firth, J., & Christensen, H. (2018). Clinical review of user engagement with mental health smartphone apps: evidence, theory and improvements. Evidence-Based Mental Health, 21(3), 116-119. doi: 10.1136/eb-2018-102891. Another clarification was that there are no pre-specified criteria for establishing the acceptability of the app or the feasibility of the trial procedures. Could the authors please outline why it was not applicable to do so? Given the CONSORT statement on randomized pilot and feasibility studies (see item 6c) I believe this clarification will strengthen the protocol. Please include a reference to justify operationalizing relapse as participant admission to an acute mental health service.
--	--

REVIEWER	Susanne Kraft District Hospital Guenzburg, Department of Psychiatry II, Ulm University, Germany
REVIEW RETURNED	15-Nov-2018

GENERAL COMMENTS	Thank you for revising the manuscript and implementing the suggested changes. I just have a few more comments: a) "Strengths and limitations of the study": "A range of types ...": As outcome measures are not of primary interest in a feasibility trial, I suggest to rephrase the sentence, e.g.: "The acceptability and feasibility of the intervention and the study design of a future full-scale RCT will be assessed via a variety of measures, including recruitment and retention rates, usage data, and qualitative interviews on experience of use." and (e.g. extra bullet point) "Established outcome measures will be used to test procedures for a future full-scale RCT and assess the estimated effects of the intervention. However, this is a feasibility trial, thus not powered ... (see last bullet point)." b) "Strengths and limitations of the study": "Assessor and participant blinding ... will not be be feasible in this trial.": As feasibility is the outcome of the planned study, this sentence is still somewhat irritating (as this is a protocol). I suggest to use another wording, e.g. "For practical reasons, there will be no blinding of assessors and participants regarding the participant's allocation (intervention vs TAU)" - or omit the sentence (seems obvious). c) p. 13/14: "The research team will have no responsibility ... would be expected to act as appropriate": Please clarify and rephrase sentences if necessary: Do you inform participants that someone will react if they state (via the app) that they feel worse? Or otherwise, do they know that this is not (necessarily) the case? Are the EIP service clinicians and carers instructed to react to the information provided by the app, or not? As this is about delivery and not development (to my understanding), please move that part to the "Delivery of the intervention" section. d) Data collection: "... participant does not pose a risk to others ...": I think an important question regarding risk is suicidality/self-harm. Thus, you might add "... and himself/herself".
--

VERSION 2 – AUTHOR RESPONSE

Reviewer(s)' Comments to Author:

Reviewer: 1

Reviewer Name: John Gleeson

Institution and Country: Australian Catholic University

Please state any competing interests or state 'None declared': None

Please leave your comments for the authors below

The authors have clarified a number of questions pertaining to the app and the measurement of outcomes which has improved the manuscript.

I have some remaining issues that I believe require some further detail based on these initial clarifications.

The authors have clarified that the researcher support will be limited to setting up participants on the app. Given the common problems of low engagement with mental health apps, could the authors please clarify their reasoning as to why this was considered sufficient for the purposes of the pilot study? In considering the response, please give consideration to the following recent review paper:

Torous, J., Nicholas, J., Larsen, M. E., Firth, J., & Christensen, H. (2018). Clinical review of user engagement with mental health smartphone apps: evidence, theory and improvements. *Evidence-Based Mental Health*, 21(3), 116-119. doi: 10.1136/eb-2018-102891.

Our response:

Thank you for your comment. We have added further details to the manuscript to outline the support participants will receive with the app. The majority of support will be provided by Early Intervention in Psychosis Service clinicians, which in line with the evidence presented in the Torous, Nicholas, Larsen, Firth & Christensen 2018 paper, may increase participant engagement with My Journey 3 (page 14).

Another clarification was that there are no pre-specified criteria for establishing the acceptability of the app or the feasibility of the trial procedures. Could the authors please outline why it was not applicable to do so? Given the CONSORT statement on randomized pilot and feasibility studies (see item 6c) I believe this clarification will strengthen the protocol.

Our response: The study features a variety of data types to explore the acceptability of the intervention, the feasibility of trial procedures and to identify potential changes to make prior to a full trial. This includes qualitative interviews, app usage data and the assessment of recruitment and retention rates. We felt that these different perspectives could not be all successfully captured by a set of criteria, and as the study is not an internal pilot specific thresholds have not been set to determine the success of the trial and its procedures.

Please include a reference to justify operationalizing relapse as participant admission to an acute mental health service.

Our response: Thank you for your suggestion. We have included a recent reference to justify the use of this for measuring relapse (page 18).

Reviewer: 3

Reviewer Name: Susanne Kraft

Institution and Country: District Hospital Guenzburg, Department of Psychiatry II, Ulm University, Germany

Please state any competing interests or state 'None declared': none

Please leave your comments for the authors below

Thank you for revising the manuscript and implementing the suggested changes. I just have a few more comments:

a) "Strengths and limitations of the study": "A range of types ...": As outcome measures are not of primary interest in a feasibility trial, I suggest to rephrase the sentence, e.g.: "The acceptability and feasibility of the intervention and the study design of a future full-scale RCT will be assessed via a variety of measures, including recruitment and retention rates, usage data, and qualitative interviews on experience of use." and (e.g. extra bullet point) "Established outcome measures will be used to test procedures for a future full-scale RCT and assess the estimated effects of the intervention. However, this is a feasibility trial, thus not powered ... (see last bullet point)."

Our response: Thank you for your suggestion. We have changed the strengths and limitation section as suggested (page 4).

b) "Strengths and limitations of the study": "Assessor and participant blinding ... will not be feasible in this trial.": As feasibility is the outcome of the planned study, this sentence is still somewhat irritating (as this is a protocol). I suggest to use another wording, e.g. "For practical reasons, there will be no blinding of assessors and participants regarding the participant's allocation (intervention vs TAU)" - or omit the sentence (seems obvious).

Our response: The sentence "Assessor and participant blinding to group allocation will not be feasible in this trial" has been omitted as suggested (page 4). Thank you for your suggestion.

c) p. 13/14: "The research team will have no responsibility ... would be expected to act as appropriate": Please clarify and rephrase sentences if necessary: Do you inform participants that someone will react if they state (via the app) that they feel worse? Or otherwise, do they know that this is not (necessarily) the case? Are the EIP service clinicians and carers instructed to react to the information provided by the app, or not? As this is about delivery and not development (to my understanding), please move that part to the "Delivery of the intervention" section.

Our response: The sentence "The research team will have no responsibility for providing clinical care if any My Journey 3 data shared by a participant to a third party indicates a decline in their mental health" has been moved to the "Delivery of the intervention" section of the methods (page 14). As suggested we have added further detail to clarify what is expected to happen if a participant shares My Journey 3 data which indicates a decline in their mental health, including that researchers, carers or clinicians may act if shared My Journey 3 data indicates a major risk but that the app is not suitable for seeking urgent medical care (page 15).

d) Data collection: "... participant does not pose a risk to others ...": I think an important question regarding risk is suicidality/self-harm. Thus, you might add "... and himself/herself".

Our response: Thank you for the suggestion. The addition to the wording of this sentence has been made (page 16).

VERSION 3 – REVIEW

REVIEWER	John Gleeson Australian Catholic University
REVIEW RETURNED	15-Jan-2019

GENERAL COMMENTS	Thank-you - this version has addressed all of my previous queries.
--

REVIEWER	Susanne Kraft District Hospital Guenzburg, Department of Psychiatry II, Ulm University, Germany
REVIEW RETURNED	08-Jan-2019

GENERAL COMMENTS	Thank you. I have no further comments.
--